# Serum miR-379 expression is related to the development and progression of hypercholesterolemia in non-alcoholic fatty liver disease

**Kinya Okamoto**[1]*, **Masahiko Koda**[1], **Toshiaki Okamoto**[1‡], **Takumi Onoyama**[1‡], **Kenichi Miyoshi**[1‡], **Manabu Kishina**[1‡], **Tomomitsu Matono**[1‡], **Jun Kato**[1‡], **Shiho Tokunaga**[1‡], **Takaaki Sugihara**[1‡], **Akira Hiramatsu**[2], **Hideyuki Hyogo**[3], **Hiroshi Tobita**[4], **Shuichi Sato**[4], **Miwa Kawanaka**[5], **Yuichi Hara**[6], **Keisuke Hino**[6], **Kazuaki Chayama**[2], **Yoshikazu Murawaki**[1], **Hajime Isomoto**[1]

1 Second Department of Internal Medicine, Tottori University School of Medicine, Yonago, Tottori, Japan, 2 Department of Gastroenterology and Metabolism, Graduate School of Biomedical and Health Sciences, Hiroshima University, Hiroshima, Hiroshima, Japan, 3 Department of Gastroenterology and Hepatology, JA Hiroshima General Hospital, Hatsukaichi, Hiroshima, Japan, 4 Department of Gastroenterology and Hepatology, Shimane University School of Medicine, Izumo, Shimane, Japan, 5 Department of General Internal Medicine 2, General Medical Center, Kawasaki Medical School, Okayama, Okayama, Japan, 6 Department of Hepatology and Pancreatology, Kawasaki Medical School, Kurashiki, Okayama, Japan

☯ These authors contributed equally to this work.
‡ These authors also contributed equally to this work.
* kinyah.okamoto@kje.biglobe.ne.jp

**Data Availability Statement:** All relevant data are within the manuscript and its Supporting Information files.

## Abstract

### Introduction

Non-alcoholic fatty liver disease (NAFLD) has a wide spectrum, eventually leading to cirrhosis and hepatic carcinogenesis. We previously reported that a series of microRNAs (miR-NAs) mapped in the 14q32.2 maternally imprinted gene region (Dlk1-Dio3 mat) are related to NAFLD development and progression in a mouse model. We examined the suitability of miR-379, a circulating Dlk1-Dio3 mat miRNA, as a human NAFLD biomarker.

### Methods

Eighty NAFLD patients were recruited for this study. miR-379 was selected from the putative Dlk1-Dio3 mat miRNA cluster because it exhibited the greatest expression difference between NAFLD and non-alcoholic steatohepatitis in our preliminary study. Real-time PCR was used to examine the expression levels of miR-379 and miR-16 as an internal control. One patient was excluded due to low RT-PCR signal.

### Results

Compared to normal controls, serum miR-379 expression was significantly up-regulated in NAFLD patients. Receiver operating characteristic curve analysis suggested that miR-379 is a suitable marker for discriminating NAFLD patients from controls, with an area under the

**Funding:** Initials of the authors who received award: MK Grant number: 26461005 The full name of funder: A grant-in-aid for scientific research category C from the Japan Society for the Promotion of Science URL of funder: websitehttp://www.jsps.go.jp/english/e-grants/index.html Did the sponsors or funders play any role in the study design, data collection and analysis, decision to publish, or preparation of the manuscript?: NO

**Competing interests:** NO authors have competing interests

curve value of 0.72. Serum miR-379 exhibited positive correlations with alkaline phosphatase, total cholesterol, low-density-lipoprotein cholesterol and non-high-density-lipoprotein cholesterol levels in patients with early stage NAFLD (Brunt fibrosis stage 0 to 1). The correlation between serum miR-379 and cholesterol levels was lost in early stage NAFLD patients treated with statins. Software-based predictions indicated that various energy metabolism–related genes, including insulin-like growth factor-1 (IGF-1) and IGF-1 receptor, are potential targets of miR-379.

## Conclusions

Serum miR-379 exhibits high potential as a biomarker for NAFLD. miR-379 appears to increase cholesterol lipotoxicity, leading to the development and progression of NAFLD, via interference with the expression of target genes, including those related to the IGF-1 signaling pathway. Our results could facilitate future research into the pathogenesis, diagnosis, and treatment of NAFLD.

## Introduction

Non-alcoholic fatty liver disease (NAFLD) is an important cause of chronic liver injury, with an increasing incidence worldwide [1]. NAFLD, regarded as a hepatic manifestation of metabolic syndrome, is defined by significant lipid deposition in hepatocytes (excessive numbers of fat-laden hepatocytes are observed by light microscopy), unrelated to excessive alcohol consumption [2]. The prevalence of NAFLD is almost 25% worldwide and expected to increase with increasing incidence of obesity and metabolic diseases such as type 2 diabetes mellitus (T2DM) and hyperlipidemia [3].

The mechanism underlying the development of NAFLD has not been fully elucidated. Currently, the multiple parallel hit theory is the most widely accepted mechanism for the progression of NAFLD [4]. This theory suggests that the disease process begins with *de novo* lipogenesis (DNL) by increase fructose consumption by western style diet and the development of insulin resistance resulting from excessive energy intake [5, 6]. Fructose is a potent lipogenic carbohydrate contributing to hepatic steatosis. Fructose is taken into hepatocyte via glucose transporter 2 and converted into fructose-1-phosphate (F1P) by fructokinase. These physiological sequences are not controlled by insulin and induced by fructose [6]. Fructose-bisphosphate aldolase B (known as hepatic aldolase) converts F1P into glycogen, glucose, lactate, and acetyl-CoA. Fructose also upregulate key transcription factors for fatty acid synthesis such as sterol response element binding protein 1c and carbohydrate responsive element binding Protein [7]. Both acetyl-CoA oversupply and induce of lipogenic enzymes increase DNL in hepatocyte strongly. Insulin resistance in turn leads to hyperinsulinemia, resulting in upregulated hepatic DNL and adipose tissue lipolysis. These "primary hits" increase the susceptibility of hepatocytes to multiple pathogenetic factors, such as upregulated expression of pro-inflammatory cytokines and eicosanoids, Fas ligand, and Toll-like receptor ligands; increased reactive oxygen species (ROS) generation; and altered production of adipokines [8]. Whole-body organs such as adipose tissue, the gut, and gut microbiota are also involved in the pathologic process [9, 10]. Collectively, these factors promote hepatocyte apoptosis through mitochondrial dysfunction [11] and an endoplasmic reticulum stress reaction [12]. Such continuous liver tissue injury ultimately leads to fibrosis [13].

The clinical status of NAFLD patients is generally classified broadly into one of just two categories: non-alcoholic fatty liver (NAFL) or non-alcoholic steatohepatitis (NASH) [14]. NAFL encompasses most of the NAFLD spectrum and is a benign condition. NASH, on the other hand, is defined as the combination of steatosis with lobular inflammation and hepatocyte ballooning; it can progress to liver fibrosis and result in cirrhosis and cancerous malignancies [14]. In contrast to NAFL, NASH is a life-threatening disease. Indeed, a cohort study showed that 35% of NASH patients die during the 7.6-year average follow-up period, whereas no NAFL patients followed in that study died during the same period [15].

Considering the wide disease spectrum of NAFLD, which can result in significant differences in prognosis, it is likely that mechanisms that regulate one or more of these multiple-hit factors exist. Some risk factors for the development of liver fibrosis in NAFLD include age over 50 years, severe obesity, complications associated with T2DM, increased ferritin levels, and patatin-like phospholipase domain–containing 3 gene polymorphisms [16, 17]. However, more-sensitive and -reliable biomarkers are urgently needed to predict outcome in NAFLD patients and enable treatment to begin in the early stage.

MicroRNAs (miRNAs) are a class of endogenous, noncoding, small RNAs that regulate gene expression [18]. Mature miRNAs are introduced into RNA-induced silencing complexes (RISCs) [19]. A RISC bearing a miRNA binds to a partially complementary mRNA sequence and represses the translation of that mRNA. Because miRNAs cause incomplete base-pair matching with mRNAs, a single miRNA can inhibit the translation of hundreds to thousands of target genes [20]. As such, miRNAs play an important role in many cellular processes, including metabolism, inflammation, and fibrosis [21]. Accumulating evidence from both animal model and human patients indicates that miRNAs contribute to the pathogenesis and progression of NAFLD. For example, the expression levels of miR-29c, miR-34a, miR-155, and miR-200b in mouse model liver and miR-122 and miR-34a in human liver are thought to be involved in the development of NASH [22–24]. Our previous study showed that a series of miRNAs mapped in the 14q32.2 maternally imprinted gene cluster region delineated by the *delta-like homolog 1* and *type III iodothyronine deiodinase* genes (Dlk1-Dio3 mat) are related to NAFLD development and progression in a NAFL/NASH mouse model (fatty liver Shionogi [FLS] and mutated leptin gene transferred FLS *ob/ob*) [25]. Seven miRNAs in the Dlk1-Dio3 mat (miR-127, -136, -376c, -379, -409-3p, -411, and -495) are strongly upregulated in both FLS and FLS *ob/ob* liver tissues. In contrast to previously reported NAFLD-related miRNAs, the expression of these seven miRNAs was higher in NAFL model mice than NASH model mice.

Recent studies have clearly indicated that miRNAs are secreted into circulating body fluids from various tissues [26]. A considerable amount of secreted miRNAs are protected from enzymatic and physical degradation by binding to proteins or lipoproteins that are then stored in exosomes [27]. These observations suggest that serum miRNAs are potential biomarkers for NAFLD, as they could reflect various pathologic changes in miRNA expression in the liver. Indeed, our preliminary study in human NAFLD patients indicated that serum levels of the respective human homologs of the candidate Dlk1-Dio3 mat miRNAs are related to NAFLD progression [25]. The aim of the present study was to examine the suitability of circulating 14q32.2 mat miRNA as a human NAFLD biomarker.

## Materials and methods

### Ethics statement

This study was approved by the committee for ethics in medical experiments on human subjects of the medical faculty of Tottori University (protocol no. 2374) and all collaborative medical institutes: Hiroshima University Hospital, JA Hiroshima General Hospital, Kawasaki

University Hospital, and Shimane University Hospital. The study was conducted in accordance with the declaration of Helsinki. Written informed consent was obtained from each patient before blood was collected.

## Sample size calculation

The effective sample size has been calculated from the outcomes of our previous clinical study about serum Dlk1-Dio3 mat miRNA expression in NAFLD patients [25]. According to the criterion of our previous study, we set up the relevant difference of miR-379 expression level is 2log2. The associated standard deviation was estimated to be 1.6log2 based on the serum miR-379 expression of normal control in our previous study [25]. Given a statistical power of 80% and 0.05 level of significance, a sample size of 11 in each group will be sufficient to detect a clinically relevant difference between groups. To allow for dropout of patients (up to 20%) and further subgroup analysis, we aim to recruit 80 participants.

## Patient recruitment and collection of blood and liver samples

Recruiting and obtaining liver tissue, blood sample and clinical data were performed at Tottori University Hospital and collaborative medical institutes: Hiroshima University Hospital, JA Hiroshima General Hospital, Kawasaki University Hospital, and Shimane University Hospital from July 2014 to March 2016. All participants were Japanese who live in the western part of Japanese main-island. NAFLD outpatients who underwent continuous clinical follow-up at Tottori University Hospital or collaborative medical institutes were recruited by interview with our study members. In interview, we gave clear exposition of the purpose, procedures, duration and potential adverse events of our study using the printed description which have been obtained the approval of the committee for ethics in medical experiments on human subjects of the medical faculty of Tottori University. Participants were given sufficient time to read the consent and have all questions answered before signing the consent form voluntary. Exclusion criteria included chronic hepatitis B or C virus infection (positive hepatitis B surface antigen or hepatitis C antibody), habitual alcohol consumption over 20 g/day, administration of liver steatotic drugs (such as glucocorticoids, tamoxifen, amiodarone, methotrexate, or valproate), primary biliary cholangitis (positive anti-mitochondrial antibody), or autoimmune liver disease (positive anti-nuclear antibody or anti-smooth muscle antibody). All NAFLD patients underwent liver biopsy to confirm the diagnoses of NAFLD, and the histologic grade and NAFLD stage was determined according to the Brunt system [28]. NAFL and NASH were defined by >5% fat-laden hepatocytes in biopsy samples and at least 6 months of continuous blood test results in which alanine aminotransferase (ALT) and aspartate aminotransferase (AST) remained at <2-fold of the normal range or in excess, respectively. Blood sample collection for serum miRNA isolation and clinical blood tests were performed at the same time and within 1 month of liver biopsy. Blood samples were collected in the fasted state. For each sample, blood serum was isolated by refrigerated centrifugation at 4˚C and $1500 \times g$ for 10 min and then stored at −80˚C until use. Eighty NAFLD patients were enrolled in this study. One NAFLD patient was excluded from this study due to low RT-PCR signal, even after 60 PCR cycles. The NAFLD patients were divided into two subgroups as follows: 9 NAFL patients, and 70 NASH patients. In another analysis, NAFLD patients were also divided into early stage (n = 53) and advanced stage (n = 26) groups. Early stage was defined as Brunt fibrosis stage 0 or 1, and the advanced stage was defined as Brunt fibrosis stage 2 to 4. As normal control, 10 outpatients with asymptomatic gallbladder stones without liver function abnormalities and fatty liver changes by ultrasound imaging were recruited. The clinicopathologic features of each patient group are shown in Table 1. We assessed whether our NAFLD sample can be

**Table 1. Clinicopathologic features of NAFLD patients and controls.**

| | Control (CON) (n = 10) | NAFL (n = 9) | NASH (n = 70) | NAFL and CON | NASH and CON | NAFL and NASH | NAFLD early stage (n = 53) | NAFLD advanced stage (n = 26) | Early stage and CON | Advanced stage and CON | Early stage and advanced stage |
|---|---|---|---|---|---|---|---|---|---|---|---|
| | | | | **p value** | | | | | **p value** | | |
| Age | 59.3 ± 16.6 | 44 ± 10 | 50 ± 16 | 0.080 | 0.162 | 0.533 | 45.4 ± 14.7 | 55.2 ± 14.9 | 0.023* | 0.742 | 0.021* |
| Gender M/F | 4 / 6 | 7 / 2 | 47 / 23 | 0.170 | 0.161 | 0.710 | 38 / 15 | 16 / 10 | 0.071 | 0.460 | 0.261 |
| BMI | 21.9 ± 5.2 | 26.4 ± 2.2 | 29.8 ± 6.3 | 0.270 | 0.002* | 0.259 | 29.8 ± 5.5 | 28.4 ± 7.2 | 0.003* | 0.024* | 0.628 |
| Brunt Stage | - | 0.89 ± 0.33 | 1.58 ± 0.87 | - | - | 0.041* | 1.0 ± 0.2 | 2.6 ± 0.6 | - | - | < 0.001* |
| Brunt Grade | - | 1.0 ± 0 | 1.58 ± 0.67 | - | - | 0.021* | 1.3 ± 0.6 | 1.9 ± 0.6 | - | - | 0.001* |
| T-Bil. | 0.8 ± 0.3 | 0.9 ± 0.3 | 1.0 ± 0.4 | 0.927 | 0.479 | 0.805 | 0.9 ± 0.4 | 1.2 ± 0.3 | 0.908 | 0.071 | 0.014 |
| Alb | 4.3 ± 0.4 | 4.6 ± 0.4 | 4.4 ± 0.4 | 0.123 | 0.175 | 0.701 | 4.5 ± 0.4 | 4.4 ± 0.4 | 0.378 | 0.880 | 0.470 |
| PT (%) | 96.7 ± 9.5 | 107.9 ± 12.5 | 99.2 ± 13.2 | 0.415 | 0.187 | 0.938 | 104.0 ± 12.6 | 92.4 ± 11.7 | 0.578 | 0.837 | 0.001* |
| AST (U/L) | 27.8 ± 18.8 | 40 ± 19 | 49 ± 19 | 0.360 | 0.005* | 0.404 | 45.5 ± 16.4 | 53.3 ± 23.1 | 0.021* | 0.002* | 0.198 |
| ALT (U/L) | 25.5 ± 15.1 | 72 ± 41 | 77 ± 40 | 0.028* | 0.001* | 0.923 | 78.3 ± 39.1 | 74.5 ± 41.6 | 0.001* | 0.002* | 0.910 |
| ALP (U/L) | 276.5 ± 91.7 | 259 ± 67 | 237 ± 84 | 0.886 | 0.350 | 0.752 | 240.5 ± 73.4 | 238.6 ± 100.2 | 0.434 | 0.451 | 0.995 |
| GGT (U/L) | 47.3 ± 45.6 | 65 ± 45 | 62 ± 45 | 0.667 | 0.598 | 0.980 | 63.7 ± 46.1 | 61.4 ± 41.6 | 0.542 | 0.676 | 0.976 |
| LDH (U/L) | 158.3 ± 45.6 | 215 ± 84 | 209 ± 47 | 0.244 | 0.226 | 0.958 | 216.3 ± 58.4 | 199.5 ± 32.6 | 0.144 | 0.391 | 0.362 |
| Ch-E (U/L) | 348.3 ± 66.2 | 351 ± 85 | 379 ± 82 | 0.997 | 0.511 | 0.634 | 388.9 ± 79.7 | 352.8 ± 84.8 | 0.310 | 0.988 | 0.150 |
| BUN (mg/dL) | 11.0 ± 2.4 | 13.8 ± 2.5 | 13.1 ± 2.4 | 0.216 | 0.301 | 0.766 | 13.1 ± 2.5 | 13.3 ± 1.9 | 0.296 | 0.276 | 0.955 |
| Cr (mg/dL) | 0.56 ± 0.17 | 0.79 ± 0.13 | 0.75 ± 0.15 | 0.054 | 0.092 | 0.638 | 0.76 ± 0.14 | 0.74 ± 0.16 | 0.068 | 0.109 | 0.920 |
| UA (mg/dL) | 5.7 ± 1.2 | 6.0 ± 1.1 | 6.3 ± 1.4 | 0.973 | 0.792 | 0.883 | 6.3 ± 1.4 | 6.2 ± 1.4 | 0.805 | 0.867 | 0.985 |
| Ferritin | 42.4 ± 33.0 | 142.1 ± 74.0 | 210.6 ± 174.5 | 0.723 | 0.338 | 0.477 | 190.6 ± 158.6 | 229.1 ± 186.6 | 0.439 | 0.287 | 0.614 |
| FBS (mg/dL) | 93.7 ± 9.7 | 104.0 ± 11.5 | 117.6 ± 45.6 | 0.849 | 0.204 | 0.621 | 117.7 ± 47.8 | 113.9 ± 33.8 | 0.220 | 0.394 | 0.923 |
| HgbA1c (%) | 6.3 ± 1.0 | 5.9 ± 0.6 | 6.3 ± 1.5 | 0.911 | 0.996 | 0.658 | 6.3 ± 1.5 | 6.2 ± 1.4 | 0.995 | 0.999 | 0.938 |
| IRI (µU/mL) | | 17.1 ± 19.6 | 18.3 ± 13.5 | - | - | 0.820 | 18.8 ± 15.0 | 17.2 ± 12.8 | - | - | 0,897 |
| HOMA-IR | | 4.6 ± 5.7 | 5.3 ± 6.7 | - | - | 0.768 | 5.5 ± 7.4 | 4.9 ± 4.5 | - | - | 0.921 |
| T-Chol (mg/dL) | 202 ± 44 | 199 ± 47 | 204 ± 35 | 0.978 | 0.988 | 0.913 | 206.6 ± 36.7 | 197.5 ± 36.3 | 0.936 | 0.940 | 0.936 |
| LDL-C (mg/dL) | 134.1 ± 37.4 | 130.3 ± 43.9 | 131.3 ± 33.2 | 0.974 | 0.978 | 0.996 | 135.1 ± 33.9 | 122.5 ± 34.9 | 0.997 | 0.709 | 0.288 |
| HDL-C (mg/dL) | 67.2 ± 34.3 | 50.9 ± 6.9 | 49.4 ± 9.0 | 0.033* | 0.004* | 0.930 | 49.1 ± 7.9 | 50.6 ± 10.6 | 0.003* | 0.012* | 0.853 |
| Non-HDL-C (mg/dL) | 150.2 ± 17.0 | 147.8 ± 47.0 | 154.1 ± 36.4 | 0.908 | 0.830 | 0.643 | 157.6 ± 5.2 | 144.5 ± 7.6 | 0.683 | 0.758 | 0.161 |
| TG (mg/dL) | 104.3 ± 64.8 | 112.1 ± 50.9 | 149.9 ± 69.0 | 0.967 | 0.139 | 0.255 | 154.5 ± 70.6 | 128.4 ± 60.7 | 0.104 | 0.629 | 0.255 |
| Compl. DM | 0 | 1 (11%) | 23 (33%) | - | - | 0.182 | 16 (30%) | 8 (31%) | - | - | 0.958 |

Value data are expressed as the mean ± standard deviation.

*: p < 0.05 in analysis of variance (ANOVA). NAFLD early stage was defined as Brunt fibrosis stage 0 or 1, and advanced stage was defined as Brunt fibrosis stage 2 to 4.
T-Bil: total bilirubin, Alb: albumin, AST: alanine aminotransferase, ALT: aspartate aminotransferase, ALP: alkaline phosphatase, GGT: gamma-glutamyl transferase, LDH: lactate dehydrogenase, Ch-E: choline esterase, BUN: blood urea nitrogen, Cr: creatinine, UA: uric acid, FBS: fasting blood sugar, HgbA1c: hemoglobin A1c which was measured by national glycohemoglobin standardization program certified method, IRI: immunoreactive insulin, HOMA IR: homeostasis model assessment of insulin resistance, T-Chol: total cholesterol, LDL-C: low-density-lipoprotein cholesterol, HDL-C: high-density-lipoprotein cholesterol, Non-HDL-C: calculated by subtracting HDL-C from T-Chol levels, TG: triacylglycerol, Compl. DM: complication of diabetes mellitus, DM can be diagnosed if the patient has HgbA1c level of 6.3% or greater on two separate blood tests.

considered representative of a larger NAFLD population. Concerning age of the participants, the mean age of our NAFLD patients are 48.6 ± 15.4. The majority of NAFLD concerning studies indicate that the median age of NAFLD patients are in the range from 40 to 59 [1]. Meta-analytic assessment of prevalence and incidence of NAFLD showed that the prevalence of NAFLD increases with age, however, the prevalence rate are similar from age 40's to 60's (26.5% to 28.9%, respectively) [1]. Former large population studies support that the prevalence of NAFLD is higher in men than in women [29]. In our present study, male NAFLD patients are also more frequently observed than female patients (54 males and 25 females). Overweight is one of the strong risk factors for NAFLD [1]. Previous Japanese NAFLD patients study also showed a clear relationship between BMI and NAFLD development [30]. In our study, mean of body mass index of NAFLD patients excess 25 (29.3 ± 6.1), which indicate overweight. Development and resolution of NAFLD were both closely related to metabolic syndrome especially diabetes mellitus (DM) [30]. Hamaguchi et al. reported that 25–40% of Japanese NAFLD patients have established DM [30]. In our study, 24 cases (30%) of NAFLD patients had complication of DM. In subgroup, 1 of NAFL (11%) and 23 of NASH (33%), and 16 of NAFLD early stage (30%) and 8 of NAFLD advanced stage (31%) had DM. Because of these demographic features, we regarded that our NAFLD participants could be considered as the representative group of larger population of NAFLD.

## miRNA expression analysis with human serum

miRNA extract from the serum, Quantitative real-time polymerase chain reaction (RT-PCR) and data analysis were carried out at Tottori University. miR-379 was selected from the putative Dlk1-Dio3 mat miRNA cluster because it exhibited the greatest difference in serum expression between NAFL and NASH patients in our preliminary study [25]. Comparing to the normal controls, serum miR-379 overexpressed in simple steatosis (we named NAFL to simple steatosis in our previous report) group (3.3 ± 3.1 log2) and down regulated in NASH patients group (-7.4 ± 5.9 log2) in our previous study [25]. A miRNeasy serum/plasma kit (Qiagen Venlo, Nederland) was used to extract miRNAs from each 200-μL serum sample according to the manufacturer's instructions. The miScript II reverse transcription kit (Qiagen) was used for reverse transcription of serum miRNA according to the manufacturer's instructions. RT-PCR was used to examine the expression levels of miRNA, and data were analyzed using the ΔΔCT method of relative quantification. Applied Biosystems TaqMan® MicroRNA Assays (Applied Biosystems, Waltham, MA, USA) and an ABI7900HT system (Applied Biosystems) were used for quantitative RT-PCR amplification of serum miRNAs. The primer sequences of hsa-miR-379 was UGGUAGACUAUGGAACGUA. We selected miR-16 as an endogenous control. miR-16 is one of the most commonly used reference miRNAs in serum miRNA expression analyses [31, 32]. To the best of our knowledge, no previous reports have indicated a relationship between liver disease and miR-16. We also examined preliminary study about RT-PCR measurement in human serum between endogenous miR-16 and non-mammal spike in control miRNA. Serum samples were obtained from normal controls (n = 10). $1.6 \times 10^8$ copies of C. elegans (Ce)-miR-39-1 (Qiagen) and 1 μg bacteriophage MS2 RNA (Roche, Penzberg, Germany) as carrier RNA were added for each 200 μL serum sample. PT-PCR was performed using the same protocol of the present study. The primer sequence of hsa-miR-16 was UAGCA GCACGUAAAUAUUGGCG. The manufacturer does not disclose the primer sequence of Ce-miR-39-1. Both of miR-16 and Ce-miR-39-1 were stably expressed between samples and their standard deviations of threshold cycles were within ± 1 cycle (34.0 ± 0.8 and 23.9 ± 0.1, respectively). We concluded that endogenous miR-16 could be applied as internal control of RT-PCR in serum miRNA.

## Predicting miRNA targets

The physiological roles of miR-379 in liver are still hardly identified. Few former studies revealed the relationship between liver function and miR-379. A comprehensive investigation for miR-379 function should be carried out. We adopted the computational miRNA target genes prediction. Software prediction of miRNA target genes is a popular and reliable method to estimate miRNA physiological functions [33]. We used web-based software DIANA microT-CDS 5.0 (http://diana.cslab.ece.ntua.gr/) for miR-379 putative target gene analysis. The threshold for the target prediction score in DIANA microT-CDS was set to 0.7. One miRNA can interference hundreds to thousands genes [20]. We selected gene ontology (GO) annotation to select NAFLD related genes. Database for Annotation, Visualization, and Integrated Discovery (DAVID) 6.8 (http://david.abcc.ncifcrf.gov/) was used for GO annotation, and the Kyoto Encyclopedia of Genes and Genomes (KEGG) was used for pathway enrichment analysis.

## Statistical analysis

Statistical analysis was performed using JMP 11.2.1 software (SAS Institute Inc., Cary, NC, USA). Value data are expressed as the mean ± standard deviation. The statistical significance of differences between groups was determined using the Student's *t* test or ANOVA, followed by Dunnett's test for multiple comparisons. Receiver operating characteristic (ROC) curve analysis was performed to assess NAFLD, NAFL, and NASH diagnostic accuracy. Linear regression analysis was used to examine correlations between miRNA levels and clinicopathologic parameters. Fisher's exact test and the chi-square test were selected depending on the sample size and used to determine distribution differences of categorical variable. Differences were considered statistically significant at a p value < 0.05.

## Results

### Serum miR-379 expression was up-regulated in NAFLD patients

Compared to controls, serum miR-379 expression was significantly up-regulated in NAFLD patients (Fig 1A). In a subgroup analysis of NAFL and NASH patients, serum miR-379 expression was significantly higher in NAFL patients than normal controls (Fig 1B). We also compared early stage NAFLD (Brunt fibrosis stage 0 to 1) and advanced-stage NAFLD (Brunt fibrosis stage 2 to 4) patients with controls. Patients with early stage NAFLD exhibited significantly higher miR-379 expression than controls (Fig 1C). Expression of miR-379 in NASH patients was also higher than in controls, but the difference was not significant (p = 0.061) (Fig 1C). There was no significant difference in miR-379 expression between NAFL and NASH patients or between those with early or advanced-stage NAFLD.

### Serum miR-379 is a potential NAFLD diagnostic marker

ROC curve analysis revealed that miR-379 is a potential marker for discriminating NAFLD patients from controls (area under the ROC curve [AUROC]: 0.72) (Fig 2). AUROC values for discriminating NAFL, NASH, and early and advanced-stage NAFLD patients from controls were 0.76, 0.72, 0.74, and 0.67, respectively (Fig 2).

**A**

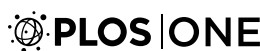

**B**

**C**

| | | | miR-379 expression (log2 folds) | p-value vs. Control |
|---|---|---|---|---|
| **A** | Control (n = 10) | | 0 ± 4.55 | 0.026* |
| | NAFLD (n = 79) | | 3.49 ± 4.58 | |
| **B** | NAFLD subgroup | NAFL (n = 9) | 4.87 ± 4.50 | 0.041* |
| | | NASH (n = 70) | 3.32 ± 4.60 | 0.061 |
| **C** | NAFLD Brunt Fibrosis stage | Early (n = 53) | 3.65 ± 4.73 | 0.039* |
| | | Advanced (n = 26) | 3.17 ± 4.35 | 0.105 |

**Fig 1. Relative expression of serum miR-379 in NAFLD patients.** A: serum miR-379 expression level between NAFLD and normal control. B: serum miR-379 expression level between NAFL, NASH. C: serum miR-379 expression level between NAFLD early stage, advanced stage and normal control. NAFLD early stage was defined as Brunt fibrosis stage 0 or 1, and advanced stage was defined as Brunt fibrosis stage 2 to 4. Value data are expressed as the mean ± standard deviation. Quantitative real-time PCR (qRT-PCR) was used to examine miRNA levels. All qRT-PCR data were normalized to that for serum miR-16, and fold-change was calculated relative to data from normal controls. *p < 0.05.

## Positive correlations were observed between serum miR-379 and alkaline phosphatase (ALP) or cholesterol levels in patients with NAFL or early stage NAFLD

We analyzed the correlations between clinicopathologic parameters and serum miR-379 levels in NAFLD patients. No significant correlation was identified between serum miR-379 expression in NAFLD patients and any of the parameters examined (S1 Fig). However, positive correlations were observed between serum miR-379 expression and ALP, total cholesterol, low-density-lipoprotein cholesterol (LDL-C) and non-HDL-C Cholesterol (non-HDL-C) levels in patients with early stage NAFLD (Fig 3). In contrast, there was no correlation between these

**B**

NAFL

AUROC: 0.76

**C**

NASH

AUROC: 0.72

**A**

NAFLD

AUROC: 0.72

**D**

NAFLD early stage

AUROC: 0.74

**E**

NAFLD advanced stage

AUROC: 0.67

**Fig 2. Receiver operating characteristic (ROC) curve analysis.** ROC curves for serum miR-379 expression level for the diagnosis of NAFLD (A) and NAFLD subgroups (B: NAFL, C: NASH, D: NAFLD early stage and E: NAFLD advanced stage). Serum miR-379 expressions were normalized to that for serum miR-16. NAFLD early stage was defined as Brunt fibrosis stage 0 or 1, and advanced stage was defined as Brunt fibrosis stage 2 to 4. AUROC means area under the ROC curve.

parameters and serum miR-379 levels in controls or patients with advanced-stage NAFLD (Fig 3, S3 Fig).

## Statin treatment weakened the correlation between miR-379 and cholesterol level

Nine of 51 patients with early stage NAFLD were undergoing treatment for hypercholesterolemia with hydroxymethyl glutaryl coenzyme A reductase (HMG CoA-reductase) inhibitors; commonly called statins. Among statin-treated and non-treated patients with early stage NAFLD, serum levels of total cholesterol, LDL-C, and triglycerides were similar (Fig 4). miR-379 expression levels were higher in the statin-treated group than the non-treated group, but the difference was not significant (5.1 ± 4.4 and 3.2 ± 4.8 log2 folds, respectively. p = 0.29). Linear regression analysis showed the non-treated group exhibited a significant positive

**NAFLD Early stage (n = 53)**

**NAFLD Advanced stage (n = 26)**

**Control (n = 10)**

### A: Total Cholesterol

$R^2 = 0.130$
**p = 0.039***

$R^2 = 0.002$
p = 0.940

$R^2 = 0.280$
p = 0.116

### B: LDL-C

$R^2 = 0.081$
**p = 0.043***

$R^2 < 0.001$
p = 0.891

$R^2 = 0.008$
p = 0.846

### C: Non-HDL-C

$R^2 = 0.082$
**p = 0.038***

$R^2 = 0.001$
p = 0.603

$R^2 = 0.002$
p = 0.924

### D: ALP

$R^2 = 0.076$
**p = 0.048***

$R^2 = 0.004$
p = 0.747

$R^2 = 0.114$
p = 0.339

**Fig 3. Correlation between miR-379 and T-Chol, LDL-C, non-HDL-C and ALP levels.** Linear regression analysis was used to examine correlations between serum miR-379 levels and clinicopathologic parameters. Left, middle, and right columns present the

results for the NAFLD early stage, NAFLD advanced stage groups and normal control group, respectively. Rows A, B, C and D show miR-379 correlation between T-Chol, LDL-C, non-HDL-C and ALP levels, respectively. $R^2$: coefficient of determination. *: $p < 0.05$. Non-HDL-C: calculated by subtracting HDL-C from T-Chol levels.

correlation between total cholesterol and serum miR-379 expression. This trend was also observed in the statin-treated group, but the correlation was not significant ($p = 0.10$) (Fig 4).

### Software-based predictions of miR-379 target genes

We predicted potential target genes of miR-379 using web-based software. Based on the selection criteria, 1423 human genes were identified as candidates. The candidate genes were classified according to GO annotation in *Homo sapiens* (Fig 5). Simple gene counting of GO terms showed that cellular process, metabolic process and biological regulation had a large proportion amount to over 70%.

| | NAFLD early stage | | |
| --- | --- | --- | --- |
| | Statin treated (n = 9) | Non-treated (n = 42) | p-value |
| T-Chol (mg/dL) | 205.4 ± 30.9 | 209.9 ± 38.1 | 0.916 |
| LDL-C (mg/dL) | 134.4 ± 32.1 | 135.3 ± 34.6 | 0.945 |
| HDL-C (mg/dL) | 50.2 ± 7.3 | 48.8 ± 8.1 | 0.631 |
| Non-HDL-C (mg/dL) | 155.2 ± 31.7 | 158.1 ± 38.7 | 0.837 |
| TG (mg/dL) | 152.1 ± 57.0 | 154.9 ± 73.6 | 0.914 |
| miR-379 (log2 fold) | 5.1 ± 4.4 | 3.2 ± 4.8 | 0.293 |

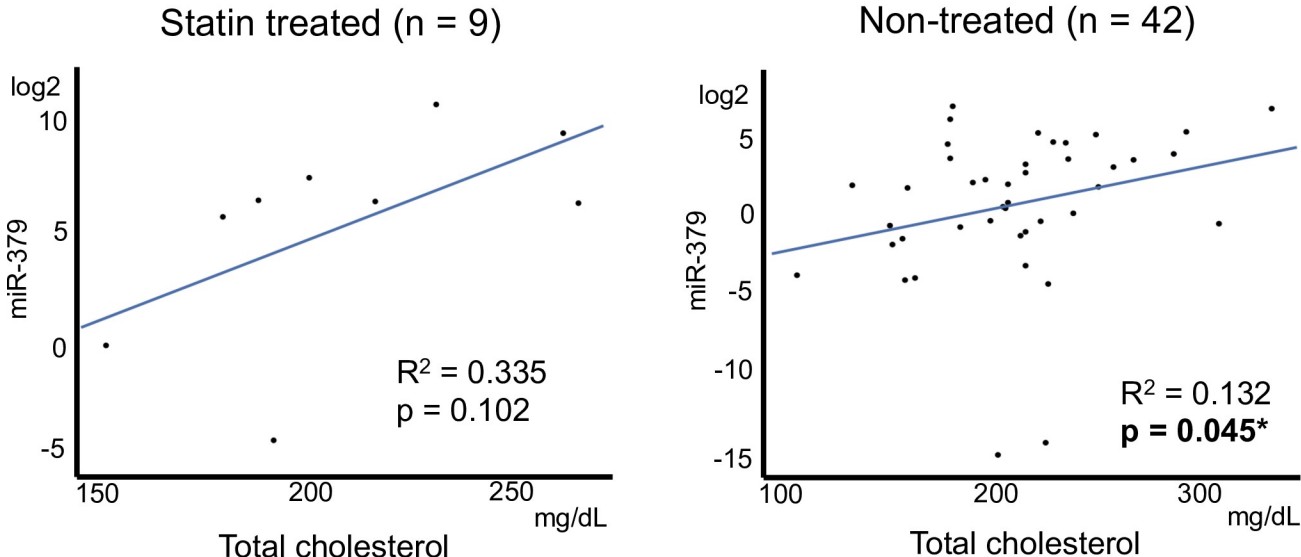

**Fig 4. Statin treatment and serum miR-379 expression, and correlation with cholesterol levels.** Value data are expressed as the mean ± standard deviation. $R^2$: coefficient of determination. *$p < 0.05$.

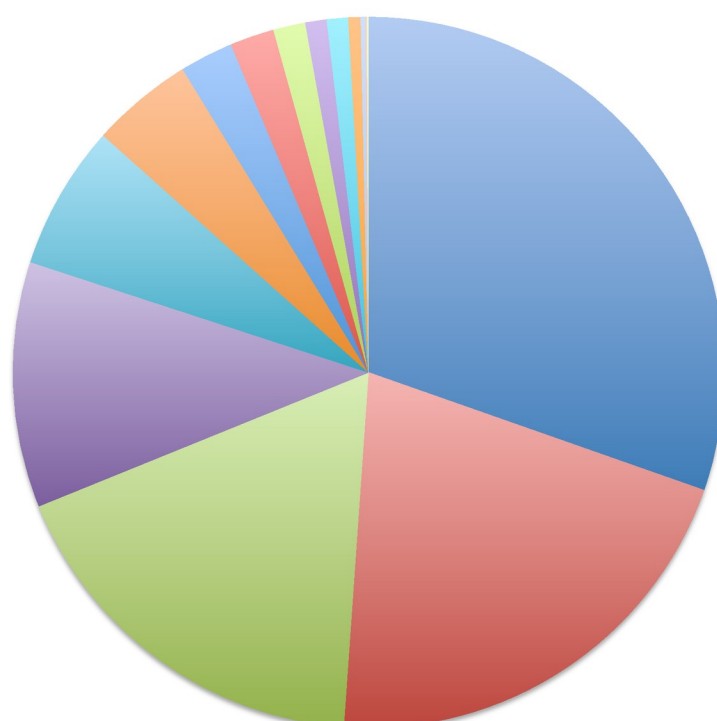

| GO term | GO ID | Gene Count | % |
|---|---|---|---|
| Cellular process | 0009987 | 438 | 31.5 |
| Metabolic process | 0008152 | 299 | 21.5 |
| Biological regulation | 0065007 | 256 | 18.4 |
| Localization | 0051179 | 161 | 11.6 |
| Multicellular organismal process | 0032501 | 95 | 6.8 |
| Response to stimulus | 0050896 | 67 | 4.8 |
| Developmental process | 0032502 | 35 | 2.5 |
| Biological adhesion | 0022610 | 29 | 2.1 |
| Immune system process | 0002376 | 21 | 1.5 |
| Cellular component organization or biogenesis | 0071840 | 14 | 1 |
| Reproduction | 0000003 | 14 | 1 |
| Cell proliferation | 0008283 | 8 | 0.6 |
| Rhythmic process | 0048511 | 3 | 0.2 |
| Biological phase | 0044848 | 1 | 0.1 |
| Pigmentation | 0043473 | 1 | 0.1 |

**Fig 5. Simple aggregation of Gene Ontology (GO) terms among putative miR-379 target genes.** The predicted miR-379 target gene dataset were fed into DAVID, version 6.8. Pie chart slices represent the number of genes associated with each GO term.

We also examined GO term enrichment analysis. The analysis can identify GO terms, which are significantly over-represented for DAVID pre-built human genome backgrounds [34]. miR-379 predicted target genes were richly represented in 12 GO terms compared to DAVID pre-built human genome backgrounds significantly (Table 2). Among 12 enriched GO, 10 terms related to cellular process regulations.

**Table 2. GO-term enrichment analysis of predicted miR-379 target genes.**

| Go Term | Gene Count | % | Fold enrichment | p value |
|---|---|---|---|---|
| Positive regulation of macromolecule biosynthetic process | 176 | 12.4 | 1.5 | > 0.001* |
| Positive regulation of RNA metabolic process | 156 | 11.0 | 1.5 | > 0.001* |
| Positive regulation of gene expression | 178 | 12.5 | 1.4 | 0.001* |
| Positive regulation of nucleobase-containing compound metabolic process | 175 | 12.3 | 1.4 | 0.001* |
| Positive regulation of cellular biosynthetic process | 181 | 12.7 | 1.4 | 0.002* |
| Positive regulation of transcription, DNA-templated | 148 | 10.4 | 1.5 | 0.002* |
| Regulation of cellular macromolecule biosynthetic process | 365 | 25.7 | 1.3 | 0.002* |
| Positive regulation of RNA biosynthetic process | 149 | 10.5 | 1.5 | 0.002* |
| Regulation of macromolecule biosynthetic process | 370 | 26.0 | 1.2 | 0.006* |
| Regulation of gene expression | 387 | 27.2 | 1.2 | 0.010* |
| Cellular protein modification process | 342 | 24.0 | 1.2 | 0.034* |
| Protein modification process | 342 | 24.0 | 1.2 | 0.034* |

Percentages indicate the number of predicted target genes associated with a GO term category compared to all predicted genes examined in the GO-term analysis. Fold-enrichment shows the abundance ratios of predicted miR-379 target genes and DAVID pre-built human genome backgrounds among GO terms. Only statistically significant results (p < 0.05) are displayed.

**Table 3. Enriched KEGG pathways among putative miR-379 target genes.**

| KEGG pathway | Gene count | % | Fold enrichment | p value |
|---|---|---|---|---|
| FOXO signaling pathway | 21 | 1.5 | 2.3 | > 0.001* |
| TGF-beta signaling pathway | 15 | 1.1 | 2.6 | 0.001* |
| Ubiquitin mediated proteolysis | 20 | 1.4 | 2.2 | 0.002* |
| Hippo signaling pathway | 19 | 1.3 | 1.8 | 0.013* |
| Prostate cancer | 13 | 0.9 | 2.2 | 0.015* |
| Transcriptional misregulation in cancer | 20 | 1.4 | 1.9 | 0.018* |
| Signaling pathways regulating pluripotency of stem cells | 17 | 1.2 | 2.3 | 0.027* |
| p53 signaling pathway | 10 | 0.7 | 1.8 | 0.036* |
| cGMP-PKG signaling pathway | 18 | 1.3 | 1.6 | 0.038* |
| Focal adhesion | 21 | 1.5 | 2.1 | 0.038* |
| mTOR signaling pathway | 9 | 0.6 | 1.7 | 0.040* |
| Melanoma | 10 | 0.7 | 2.2 | 0.048* |

Percentages indicate the number of predicted miR-379 target genes associated with a KEGG pathway compared to all predicted genes explored in the KEGG pathway analysis. Fold-enrichment shows the abundance ratios of predicted miR-379 target genes and DAVID pre-built human genome backgrounds among KEGG terms. Only statistically significant results ($p < 0.05$) are displayed.

Each of the enriched GO terms still had hundreds genes (Table 2). It is one of the common weak points of this method that the output of genes can be large [34]. To select putative target gene for NAFLD pathology, we examined extending analysis using backend annotation database. Our GO term analysis showed that miR-379 seemed to relate biological regulations largely (Table 2). Therefore we selected KEGG pathway as the backend database. Ontology annotation via KEGG pathway mapping showed that biological functions have been identified for 32.8% of the candidate genes (467 of 1423 genes). Function-labeled miR-379 candidate target genes were primarily enriched in clusters associated with nutrition and energy regulation (FOXO and mTOR signaling pathways), cancer (melanoma, prostate cancer, p53 signaling, Hippo signaling, and transcriptional misregulation in cancer), and multi-functional cellular mechanisms or signaling pathways (cGMP-PKG signaling, focal adhesion, Hippo signaling pathway, pluripotency regulation in stem cells, TGF-beta signaling, and ubiquitin-mediated proteolysis) (Table 3).

Finally, to identify probable miR-379 target genes related to the pathology of NAFLD, we conducted a keyword search of the U.S. National Library of Medicine database PubMed (https://www.ncbi.nlm.nih.gov/pubmed) using the terms "KEGG annotated putative target gene" and "NAFLD" or "NASH". A total of 27 predicted genes were associated with NAFLD development or progression, including multi-functional cellular mechanisms or signaling pathways (*HDAC2*), fibrosis and inflammation (*CAT*, *CTGF*, *IL10*, *PDGFA*, *PDGFRA*, *SMAD4*, *TGFBR1*, and *THBS1*), cell survival and proliferation (*Bcl2*, *CCNB1*, *HGF*, *PMAIP1*, *PTEN*, and *YAP1*), and energy management, including gluconeogenesis and lipogenesis (*CREB1*, *EIF4E*, *FOXO1*, *INSR*, *IGF1*, *IGF1R*, *ITPR2*, *PRKAA1* and *2*, *RICTOR*, *SOCS1*, and *TCF7L2*) (Table 4) [35–60].

## Discussion

The present study revealed significantly higher serum levels of miR-379 in NAFLD patients compared to controls. Our previous study indicated that miR-379 expression in liver tissues of an NAFLD mouse model is strongly upregulated (>4 log2 compared to the normal control group) [25]. miR-379 secretion from liver tissue, probably via exosome particles rich in miR-

**Table 4. Keyword search of the U.S. National Library of Medicine database PubMed to identify KEGG annotated miR-379 putative target genes associated with NAFLD or NASH.**

| Gene Code | Protein name | Reference |
|---|---|---|
| Bcl2 | *B-cell lymphoma 2* | Panasiuk et al. 2006 |
| CAT | *Catalase* | Kumar et al. 2013 |
| CCNB1 | *Cyclin B1* | Gentric et al. 2015 |
| CREB1 | *cAMP responsive element binding protein 1* | Oh et al. 2013 |
| CTGF | *Connective tissue growth factor* | Colak et al. 2012 |
| EIF4E | *Eukaryotic translation initiation factor 4E* | Wang et al. 2014 |
| FOXO1 | *Forkhead box o1* | Pan et al. 2017 |
| HDAC2 | *Histone deacetylase 2* | Kolodziejczyk et al. 2019 |
| HGF | *Hepatocyte growth factor* | Kosone et al. 2007 |
| INSR | *Insulin receptor* | Wu et al. 2017 |
| IGF1 | *Insulin like growth factor 1* | Adamek et al. 2018 |
| IGF1R | *Insulin like growth factor 1 receptor* | Go et al. 2014 |
| IL10 | *Interleukin 10* | Cintra et al. 2008 |
| ITPR2 | *Inositol 1, 4, 5-trisphosphate receptor type 2* | Khamphaya et al. 2018 |
| PDGFA | *Platelet derived growth factor subunit A* | Hardy et al. 2017 |
| PDGFRA | *Platelet derived growth factor receptor A* | Abderrahmani et al. |
| PMAIP1 | *Phorbol 12-myristate 13-acetate induced protein 1* | Kung et al. 2016 |
| PRKAA1 | *5' AMP-activated protein kinase catalytic subunit alpha 1* | Garcia et al. 2019 |
| PRKAA2 | *5' AMP-activated protein kinase catalytic subunit alpha 2* | Garcia et al. 2019 |
| PTEN | *Phosphatase and tensin homolog* | Matsuda et al. 2013 |
| RICTOR | *Rapamycin-insensitive companion of mammalian target of rapamycin* | Sydor et al. 2017 |
| SMAD4 | *Small worm phenotype and mothers against decapentaplegic 4* | Qin et al. 2018 |
| SOCS1 | *Suppressor of cytokine signaling 1* | Wang et al. 2017 |
| TCF7L2 | *Transcription factor 7-like 2* | Musso et al. 2009 |
| TGFBR1 | *Transforming growth factor beta receptor 1* | Matsubara et al. 2012 |
| THBS1 | *Thrombospondin 1* | Li et al. 2017 |
| YAP1 | *yes-associated protein 1* | Chen et al. 2018 |

379, appears to be related, at least in part, to the high circulating level observed in NAFLD patients.

Relatively little is known regarding the mechanism regulating miR-379 expression. miR-379 has been mapped to the miRNA cluster in the Dlk1-Dio3 mat region. Major regulators of Dlk1-Dio3 locus expression include methylated regulatory regions such as the germline-derived intergenic differentially methylated region and somatic MEG3-differentially methylated region [61, 62]. Moreover, CpG islands that are embedded in or near miRNA-coding regions also regulate the expression of Dlk1-Dio3 mat miRNA [63]. Dai et al. reported that miR-379 expression is directly regulated by DNA methylation [64]. In addition, histone acetylation functions synergistically with DNA methylation to regulate the Dlk1-Dio3 locus [63].

With respect to non–DNA methylation regulation, Guia and colleagues reported that the miRNA cluster miR-379/410 is a direct transcriptional target of the glucocorticoid receptor, which promotes insulin resistance and systemic dyslipidemia [65]. Guia et al. also showed that miR-379 is upregulated in liver tissue of obese subjects and that hepatic miR-379 expression in patients with obesity is correlated with both serum cortisol and triacylglycerol (TG) levels [65]. However, in our present study, TG levels in NAFLD patients did not differ significantly from those of controls (Table 1), and serum miR-379 expression was not correlated with TG level (p = 0.738, S1 Fig). This discrepancy may be related to differences between obese patients and

NAFLD patients whose diagnosis was confirmed by liver biopsy. The mechanism of serum miRNA expression may also be related to this discrepancy. For example, sorting and selection occur during incorporation of cytosolic miRNAs into exosomes [66]. Because the level of circulating miRNAs is the sum total of miRNAs secreted from tissues/organs throughout the body, other metabolism-related organs may affect the level of circulating miRNA. Chartoumpekis et al. reported that miR-379 is overexpressed in white adipose tissue in an obese mouse model [67].

ROC curve analyses showed that miR-379 provides fair diagnostic accuracy for NAFLD. The AUROC of serum miR-379 for NAFLD diagnosis was >0.7 and similar to other single serologic markers for non-invasive detection of NAFLD, such as tumor necrosis factor–alpha, interleukin-6, and ferritin [68]. Most non-invasive NAFLD markers exhibit higher values and diagnostic accuracy in patients with liver fibrosis and cirrhosis [69]. In contrast to the majority of NAFLD diagnostic markers, the serum miR-379 level was significantly increased relative to NAFL, but there was no difference between NAFL and NASH. This distinctive feature of serum miR-379 may confer an advantage for detecting NAFLD in the early stage. For instance, serum miR-379 is a candidate factor for use in NAFLD diagnosis algorithms combining multiple biomarkers as a means of increasing sensitivity for early stage diagnosis [70].

Our present study showed that the serum miR-379 level is positively correlated with ALP in early stage NAFLD. Serum ALP is the traditional marker of cholestasis. However, the other cholestasis markers, such as bilirubin and gamma-glutamyl transferase, were not significantly correlated with miR-379 (S2 Fig). ALP is a plasma membrane–bound enzyme that catalyzes the hydrolysis of phosphate esters [71]. Though found in most body tissues, ALP is particularly abundant in the liver, bone, kidneys, and intestinal mucosa, with liver and bone serving as the predominate organs supplying ALP to circulating body fluids [71]. Chronic liver diseases, including NAFLD, increase serum ALP levels [72, 73]. Moreover, previous reports indicated that the serum ALP level is an independent marker of NAFLD development and progression. Pantsari et al. showed that a subset of NAFLD patients (elderly females) exhibit isolated elevation in ALP rather than aminotransferases [74]. Kocabay et al. reported that serum levels of ALP, but not gamma-glutamyl transferase, are increased in NAFLD patients with early fibrosis stage (stage 1 to 2) [75]. ALP is richly expressed in the canalicular membrane side of hepatocytes, and previous studies suggested that ALP relates the transport of bile acid, which plays a major role in cholesterol metabolism and excretion [76]. However, details regarding the physiologic functions of ALP are unclear. miR-379 may be related to NAFLD development and progression by directly or indirectly modulating ALP expression.

Our present study also showed that the serum miR-379 level is positively correlated with serum cholesterol in early stage NAFLD. The contribution of hypercholesterolemia to the development of NAFLD has not been fully elucidated; however, previous studies showed that hepatic cholesterol synthesis and circulating total cholesterol and LDL are increased in NAFLD [77]. Disruption of hepatic cholesterol homeostasis and free cholesterol (FC) accumulation in liver tissue is related to the pathogenesis of NAFLD [78, 79]. Some studies have shown that hepatic cholesterol synthesis is up-regulated in NAFL and NASH patients due to increased activity of a major regulator of cholesterol synthesis, sterol regulatory element–binding protein 2 and its downstream effector HMG CoA-reductase, which catalyzes a rate-limiting step in cholesterol synthesis [80–82]. Interestingly, Min et al. also reported that up-regulation of cholesterol synthesis was not observed in control obese subjects [80].

Regarding other cholesterol-related metabolic functions in the liver of NAFLD patients, cholesterol de-esterification is increased, and cholesterol catabolism to bile acid and cholesterol efflux via the bile duct are attenuated [80]. These NAFLD-specific changes in cholesterol metabolism are believed to increase FC levels in liver tissues. FC accumulation in hepatocytes

induces mitochondrial dysfunction, which results in increased production of ROS and leads to the unfolded protein response in the endoplasmic reticulum, leading to localized stress and apoptosis [79]. Mari et al. also reported that FC loading (but not that of fatty acids or triglycerides) into hepatocyte mitochondria membranes sensitizes the hepatocyte to pro-inflammatory cytokines (e.g., tumor necrosis factor–alpha and Fas) in mouse models, resulting in steatohepatitis [83]. Moreover, FC accumulation in non-parenchymal cells in liver tissues such as Kupffer cells and stellate cells promotes activation of these cells [84, 85]. The activated Kupffer cells secrete pro-inflammatory cytokines such as interleukin-1β and tumor necrosis factor– alpha, and activated stellate cells differentiate into myofibroblasts, which exhibit a high ability to produce extracellular matrix and fibrogenic cytokines, such as transforming growth factor– β [84, 85]. It has been hypothesized that miR-379 promotes the development and progression of NAFLD as a result of continuous over-nutrition—manifested primarily as obesity—by increasing the lipotoxicity of cholesterol. Cirrhosis and hepatocellular carcinoma are the most common liver-related causes of morbidity associated with NAFLD [86]. However, cardiovascular disease is the most common cause of death in NAFLD patients without cirrhosis [15]. Therefore, some reviewers have recommended giving priority to the prevention of cardiovascular or renal diseases over liver-specific treatments in patients with non-aggressive NAFLD [87].

miR-379 has also been associated with the risk of cardiovascular disease in early stage NAFLD via up-regulation of the serum cholesterol level, which plays an important role in atherosclerosis development. In the present study, however, no significant correlation between serum miR-379 and cholesterol levels was observed in control subjects and NAFLD patients with advanced fibrosis (Brunt stage 2 to 4). This suggests that such a correlation is pertinent only under limited conditions, such as early stage NAFLD–specific pathophysiologic and nutritional states. The serum miR-379 level in controls was significantly lower than that in patients with early stage NAFLD. Normal levels of miR-379 may be insufficient to affect cholesterol metabolism. With respect to advanced-stage NAFLD, it is known that serum cholesterol levels decline with progression of liver fibrosis, independent of the etiology of chronic liver disease [88]. The effect of miR-379 on cholesterol metabolism may be attenuated by decreased hepatic parenchymal function.

The present study also demonstrated that the use of statins to treat hypercholesterolemia in NAFLD patients weakens the relationship between serum miR-379 and cholesterol levels. Statins target hepatocytes and inhibit HMG-CoA reductase, which catalyzes the rate-limiting step in the cholesterol biosynthesis pathway, known as the mevalonate pathway [89]. HMG-CoA reductase converts HMG-CoA into mevalonic acid, a cholesterol precursor. Stains have a higher binding affinity for HMG-CoA reductase than HMG-CoA and thus block access to the active site by the substrate [89]. Previous studies indicated that statins improve hepatic steatosis and reduce hepatic inflammation and fibrosis in NAFLD patients [90, 91]. Moreover, long-term observations of NAFLD patients indicated that continuous statin treatment reduces rates of liver-related death and liver transplantation [92]. Statins may attenuate the effect of miR-379 on cholesterol biosynthesis, resulting in reduced cholesterol lipotoxicity in NAFLD.

GO term annotation analyses showed enrichment of cellular biosynthesis and metabolism– related genes among predicted miR-379 targets. Aberrations in biosynthesis and metabolism play important roles in metabolic disorders such as NAFLD. miR-379 appears to affect the development and progression of NAFLD by interfering with these target genes.

KEGG pathway mapping of prospective miR-379 target genes extracted biological functions such as nutrition and energy regulation, the down-regulation of which leads to the development of NAFLD. Searches of PubMed combining keywords with the selected putative target genes identified in the KEGG pathway analysis and NAFLD identified a number of

metabolism-, inflammation-, and fibrosis-related genes. Among the selected putative target genes, *IGF1* and *IGF1R* were identified as targets of miR-379 interference in previous studies [93, 94]. IGF-1 is an insulin-like anabolic hormone primarily secreted by hepatocytes, and circulating IGF-1 levels reflect hepatic IGF-1 expression [95]. Previous studies reported that adults with growth hormone deficiency in which hepatic IGF-1 production is impaired exhibit an increased prevalence of NASH; IGF-1 substitution ameliorated NAFLD in a mouse model [96, 97]. In NAFLD patients without growth hormone deficiency, serum IGF-1 levels are also significantly reduced [95, 98]. The mechanism by which IGF-1 and its signaling pathways protect against NAFLD have been found to involve a variety of biological functions, such as improving insulin sensitivity, decreasing ROS production, and inducing senescence of hepatic stellate cells [99–101]. With respect to lipid metabolism, it has been reported that IGF-1 accelerates lipid oxidation and lipolysis [99]. Moreover, several previous studies revealed that serum IGF-1 is inversely correlated with serum levels of total cholesterol and LDL-C [102]. *IGF1* appears to be one of the most significant miR-379 target genes with regard to promoting the development and progression of NAFLD via the enhancement of cholesterol lipotoxicity. Among other keyword-selected putative target genes, B-cell lymphoma 2 (*BCL2*), catalase (*CAT*), and cAMP responsive element binding protein 1 (*CREB1*) are reportedly down-regulated in the liver in NAFLD [36, 103, 104]. *BCL2* and *CAT* are major anti-apoptosis genes that function by protecting against mitochondrial outer membrane permeabilization and detoxifying ROS, respectively [36, 103]. Down-regulation of *BCL2* and *CAT* expression in liver tissue drives hepatocyte apoptosis, which is an important pathologic event in the development and progression of NAFLD. CREB1 is a transcription factor that regulates energy balance by suppressing hepatic fatty acid generation and accumulation via downregulation of hepatic-specific peroxisome proliferator activated receptor–γ and fatty acid transporter CD36 expression [104]. miR-379 may affect the development and progression of NAFLD by interfering with the expression of these target genes, which is reportedly down-regulated in NAFLD.

A relationship with NAFLD has also been reported for other miR-379 target genes. For example, 5'-AMP–activated protein kinase catalytic subunit alpha 2 (*PRKAA2*) is the catalytic subunit alpha 2 of AMPK, a key sensor of energy status in mammalian cells. In the liver, AMPK phosphorylates and inactivates both acetyl-coenzyme A carboxylase and HMG-CoA reductase [105]. Acetyl-coenzyme A carboxylase regulates the biosynthesis of malonyl-CoA, which is the initial committed intermediate in fatty acid biosynthesis. Malonyl-CoA can inhibit carnitine palmitoyl transferase 1, which controls mitochondrial fatty acid oxidation [106]. Therefore, AMPK downregulation increases fatty acid and cholesterol biosynthesis and inhibits fatty acid oxidation, resulting in hepatic lipid accumulation. Although AMPK appears to be related to NAFLD development, details regarding levels of AMPK in hepatocytes are controversial [107].

Previous studies reported the relationship between miR-379 and various diseases. The majority of these studies suggest that miR-379 plays tumor suppressive role in many types of carcinomas, including nasopharyngeal carcinoma, cervical cancer, lung cancer, gastric cancer, hepatocellular carcinoma, bladder cancer, and osteosarcoma [108–113]. With regard to metabolic disorders as described above, de Guia et al. revealed a relationship between miR-379 and lipid homeostasis dysregulation [65]. Additionally, patients with a congenital disease known as maternal uniparental disomy for chromosome 14, which causes overexpression of miR-379 of the Dlk1-Dio3 mat miRNA cluster, exhibit characteristic weight gain in early childhood that results in truncal obesity [114].

Our study had some limitations associated with sample size and study design. We did not perform spike in control measurement in NAFLD patients. We cannot assess equalities in RNA extraction efficacies and RT-PCR accurately measurement between NAFLD serum

samples especially the drop out case. We used software programs to predict target genes of the candidate miRNAs. Although this method is commonly used, it carries a risk of missing some real targets because the software is designed to assess the relative strength of partial sequence complementarity between mRNA and miRNA. Ontology selection was used to select putative targets that might be relevant to cellular functions. However, ontology selection can only identify proteins for which the function has been identified. Notably, our understanding of the detailed mechanisms that promote the development and progression of NAFLD to NASH is still developing, but new insights are being obtained regularly.

Moreover, we did not confirm whether miR-379 actually interfered with any of the predicted target genes in vivo (e. g. expression measurement in serum or liver tissue) or in vitro, such as direct binding experiments or miR-379 ectopic overexpression by gene transfection. Complex intracellular regulatory networks influence the tissue-specific function of miRNAs [115]. Therefore, further studies are needed to assess whether the predicted targets are actual targets of miR-379 in NAFLD.

Concerning the correlation between serum ALP and miR-379, we could not definitively conclude that the correlation reflects only liver tissue pathologic changes. Bone is another major ALP-secreting organ, and the serum level of the bone isozyme of ALP is elevated in children, adolescents, and elderly people due to bone tissue turnover [116, 117]. Regarding our study participants, all NAFLD patients and control subjects were adults (age ranging from 20 to 76 years), and there was no significant relationship between serum ALP level and age ($R^2 = 0.0286$; $p = 0.115$). Additionally, no pregnant subjects were included. The number of patients in this study was small, at less than 100. Consequently, the statistical power of the human serum data was relatively limited.

In conclusion, the serum level of miR-379, a member of Dlk1-Dio3 mat miRNA cluster, exhibits high potential as a biomarker for NAFLD. miR-379 also appears to increase cholesterol lipotoxicity, which promotes the development and progression of NAFLD by interfering with the expression of target genes, including those of the IGF-1 signaling pathway. To confidently identify more associations between highly complex and interactive miRNAs with NAFLD, future longitudinal studies with greater sample sizes will be necessary.

## Supporting information

**S1 Fig. Linear regression analysis of relationships between serum miR-379 and clinical features of NAFLD patients.** Normalized relative to serum miR-16; miR-379 values represent fold-difference relative to the normal control.
(TIF)

**S2 Fig. Linear regression analysis of the relationships between serum miR-379 and clinical features of early stage NAFLD patients (Brunt fibrosis stage 0 to 1).** Normalized relative to serum miR-16; miR-379 values represent fold-difference relative to the normal control.
(TIF)

**S3 Fig. Linear regression analysis of the relationships between serum miR-379 and clinical features of advanced-stage NAFLD patients (Brunt fibrosis stage 2 to 4).** Normalized relative to serum miR-16; miR-379 values represent fold-difference relative to the normal control.
(TIF)

## Author Contributions

**Conceptualization:** Kinya Okamoto, Masahiko Koda, Yoshikazu Murawaki.

**Data curation:** Kinya Okamoto, Toshiaki Okamoto, Takumi Onoyama, Kenichi Miyoshi, Manabu Kishina, Tomomitsu Matono, Jun Kato, Shiho Tokunaga, Takaaki Sugihara, Akira Hiramatsu, Hideyuki Hyogo, Hiroshi Tobita, Shuichi Sato, Miwa Kawanaka, Yuichi Hara.

**Formal analysis:** Kinya Okamoto.

**Funding acquisition:** Masahiko Koda.

**Investigation:** Kinya Okamoto, Masahiko Koda, Toshiaki Okamoto, Takumi Onoyama, Kenichi Miyoshi, Manabu Kishina, Tomomitsu Matono, Jun Kato, Shiho Tokunaga, Takaaki Sugihara, Akira Hiramatsu, Hideyuki Hyogo, Hiroshi Tobita, Shuichi Sato, Miwa Kawanaka, Yuichi Hara.

**Methodology:** Kinya Okamoto.

**Project administration:** Kinya Okamoto, Hajime Isomoto.

**Resources:** Kinya Okamoto, Masahiko Koda, Toshiaki Okamoto, Kenichi Miyoshi, Manabu Kishina, Tomomitsu Matono.

**Software:** Kinya Okamoto.

**Supervision:** Kinya Okamoto, Masahiko Koda, Keisuke Hino, Kazuaki Chayama, Yoshikazu Murawaki, Hajime Isomoto.

**Validation:** Kinya Okamoto.

**Visualization:** Kinya Okamoto.

**Writing – original draft:** Kinya Okamoto.

**Writing – review & editing:** Kinya Okamoto, Masahiko Koda.

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
