## [Decision Letter · Decision Letter 0]

21 Nov 2019

PONE-D-19-17509

Serum miR-379 expression is related to the development and progression of hypercholesterolemia in non-alcoholic fatty liver disease

PLOS ONE

Dear Dr, Okamoto

Thank you for submitting your manuscript to PLOS ONE. After careful consideration, we feel that it has merit but does not fully meet PLOS ONE’s publication criteria as it currently stands. Therefore, we invite you to submit a revised version of the manuscript that addresses the points raised during the review process.

More specifically, you need to confirm the level of expression of miR-379 target genes.

We would appreciate receiving your revised manuscript by the 20th of February 2020. To enhance the reproducibility of your results, we recommend that if applicable you deposit your laboratory protocols in protocols.io, where a protocol can be assigned its own identifier (DOI) such that it can be cited independently in the future. For instructions see: http://journals.plos.org/plosone/s/submission-guidelines#loc-laboratory-protocols

We look forward to receiving your revised manuscript.

Kind regards,

Catherine Mounier

Academic Editor

PLOS ONE

Journal Requirements:

2. In your Methods section, please provide additional information about the participant recruitment method and the demographic details of your participants. Please ensure you have provided sufficient details to replicate the analyses such as: a) the recruitment date range (month and year), b) a description of any inclusion/exclusion criteria that were applied to participant recruitment, c) a statement as to whether your sample can be considered representative of a larger population, d) a description of how participants were recruited, and e) descriptions of where participants were recruited and where the research took place.

In your Methods section, please provide a sample size calculation.

Reviewers' comments:

Reviewer's Responses to Questions

**Comments to the Author**

1. Is the manuscript technically sound, and do the data support the conclusions?

Reviewer #1: Yes

Reviewer #2: Yes

Reviewer #3: Partly

2. Has the statistical analysis been performed appropriately and rigorously? 

Reviewer #1: Yes

Reviewer #2: Yes

Reviewer #3: I Don't Know

3. Have the authors made all data underlying the findings in their manuscript fully available?

Reviewer #1: Yes

Reviewer #2: Yes

Reviewer #3: Yes

4. Is the manuscript presented in an intelligible fashion and written in standard English?

Reviewer #1: Yes

Reviewer #2: Yes

Reviewer #3: No

5. Review Comments to the Author

Reviewer #1: Okamoto and colleagues set out to examine the suitability of circulating serum miR-379 as a potential non-invasive biomarker for non-alcoholic fatty liver disease using real-time PCR platform. The manuscript was well-written and has merit. However, there are some points that need to be clarified.

1. The presence of inhibitors in serum samples may limit the ability to extract RNA or accurately measure miRNAs by real-time PCR. I was wondering whether a spiked-in control was also included in this study to ensure efficiency of RNA extraction?

2. It is not clear how reverse transcription was carried out prior to real-time PCR. Please refer to lines 175-178.

3. It would be interesting to find out whether non-HDL cholesterol (total cholesterol minus HDL cholesterol) is a predictor for non-alcoholic fatty liver disease. Please refer to Table 1, page 11.

4. What was the basis of choosing miR-379 from a series of potential miRNAs (miR-127, miR-136, miR-376c, miR-379, miR-409-3p, miR-411, miR-495) mapped in the 14q32.3 maternally imprinted gene region? Please refer to lines 29-33 and lines 105-111).

Reviewer #2: The manuscript of Okamoto and colleagues highlights the correlation between the expression of miR-379 and NASH development and progression. The study was performed with serum collected from patients affected by early and advanced NASH.

The study was performed in an extremely accurate manner and based its achievement on the correlation between expression of miR-379 and the clinical markers of NASH, especially cholesterol lipotoxicity.

The soundness of the study is supported by previous study published from the same groupd with NASH mouse model.

Some minor points need to be addressed:

The number of patients involved in the study should be corrected between the abstract, the methods and the results section. Please include in the Abstract the number of control samples included in the study. In the Methods section is stated that the patients are 90 and it does not correspond with the abstract. In the Results section, the authors explain that one patient was excluded from the study. This information should be included in the Methods.

In order to better follow the results described in the text, to add capital letters to differentiate the different panels of the figures when necessary. For example, Figure 1 showing three different graphs of different patient groups.

Reviewer #3: The authors report increased expression of miR-379 in patients with NAFLD as compared to 10 control patients. They further show that miR-379 may be used as marker of NAFLD with AUROC of 0.76. Computer based prediction of miR-379 targets was preformed and IGF1, IGF1R and INSR were suggested as its targets. These results are of potential interest as they are reporting on potential biomarker of NAFLD in human subjects. Unfortunately, the authors did not adequately describe their results and the Result section appears much shorter than the Introduction. Also, they did not make an attempt to validate downstream targets of miR-379, which is a major missed opportunity.

Major Remarks

It is unclear how many patients were enrolled in the study. The authors state that “Ninety patients were enrolled in this study. The patients were divided into three groups, as follows: 10 patients with asymptomatic gallbladder stones as disease controls, 9 NAFL patients, and 71 NASH patients. In another analysis, NAFLD patients were divided into early stage (n = 53) and advanced-stage (n = 26) groups. (Lane 135-137).” Was the second analysis done on the same set of pts, or was it done on NASH pts only reported in the first analysis. If it was done on NASH pts why are the number different? First cohort 71 NASH pts, while second analysis included 53+26 = 79 patients. To make this clearer, table one should list the number of pts in each group. Furthermore, it should be clearly stated if this was done on one group of pts with two separate analysis or if two cohorts of pts were used.

I could not readily find figure legends. Some description of figures was found in the Results section but it was short and limited. It did not sufficiently explain the figures, nor did it indicate how many pts were included in each group. Is +/- denoting SE or SD? Figure 2 has only title listed and no description of the figure at all!

It is important to better understand the role of miR-379 in liver physiology. The authors attempted to do this by identifying potential target genes of miR-379 using web-based software. More information needs to be provided about this process before we can make any conclusions about the results that they are showing us.

The authors found 27 predicted gene targets of miR-379 that were associated with NAFLD development or progression. They focused their discussion on insulin-like growth factor-1 (IGF-1) and IGF-1 receptor. It would be imperative to verify expression of IGF1, IGF1R and INSR in these patients and determine if their expression indeed correlates with expression of miR-379.

Minor remarks:

That authors state that NAFLD “disease process begins with the development of insulin resistance resulting from excessive energy intake [5]. (Lanes 66-67). While insulin resistance may be universally found in pts with NAFLD, its role in pathogenesis is still unclear. Competing theories exist, with one suggesting that hepatic insulin resistance initiates development of NAFLD, whereas the other argues that insulin resistance is merely a consequence of increased liver lipid deposition. If the authors choose to stand by their wording, a better reference is needed than #5, which mainly talks about the effects of fructose in pathogenesis of NAFLD.

6. PLOS authors have the option to publish the peer review history of their article (what does this mean?). If published, this will include your full peer review and any attached files.

Reviewer #1: Yes: Michael O. Baclig

Reviewer #2: Yes: Pietro Di Fazio

Reviewer #3: Yes: Samir Softic

---

## [Author Response · Author response to Decision Letter 0]

27 Jan 2020

Dear academic editor Catherine Mounier,

We greatly appreciate the opportunity to revise our manuscript entitled “Serum miR-379 expression is related to the development and progression of hypercholesterolemia in non-alcoholic fatty liver disease” (ID: PONE-D-19-17509). 

We also thank the reviewers very much for their careful reading our manuscript and insightful comments. We revised our manuscript according to journal requirements and reviewers’ comments. 

Our alterations as a result of the academic editor and the reviewer’s comments are as follows. 

Sincerely,

Kinya Okamoto

Second Department of Internal Medicine

Tottori University School of Medicine

E-mail: kinyah.okamoto@kje.biglobe.ne.jp

Journal Requirements:

Reply: We revised our manuscript style in accordance with PLOS ONE style template.

2. In your Methods section, please provide additional information about the participant recruitment method and the demographic details of your participants. Please ensure you have provided sufficient details to replicate the analyses such as: a) the recruitment date range (month and year), b) a description of any inclusion/exclusion criteria that were applied to participant recruitment, c) a statement as to whether your sample can be considered representative of a larger population, d) a description of how participants were recruited, and e) descriptions of where participants were recruited and where the research took place.

Reply: According to advices of the academic editor, we revised materials and methods part in our manuscript. 

a) The recruitment date range

Reply: We indicated the date range (from July 2014 to March 2016).

b) A description of any inclusion/exclusion criteria that were applied to participant recruitment 

Reply: We revised inclusion/exclusion criteria to enter into detail. 

c) A statement as to whether your sample can be considered representative of a larger population 

Reply: We described the assessment of representativeness of our NAFLD participants as follows.

(Lines 231-250 in revised manuscript with track change)

We assessed whether our NAFLD sample can be considered representative of a larger NAFLD population. Concerning age of the participants, the mean age of our NAFLD patients are 48.6 ± 15.4. The majority of NAFLD concerning studies indicate that the median age of NAFLD patients are in the range from 40 to 59 (1). Meta-analytic assessment of prevalence and incidence of NAFLD showed that the prevalence of NAFLD increases with age, however, the prevalence rate are similar from age 40’s to 60’s (26.5 % to 28.9 %, respectively) (1). Former large population studies support that the prevalence of NAFLD is higher in men than in women (29). In our present study, male NAFLD patients are also more frequently observed than female patients (54 males and 25 females). Overweight is one of the strong risk factors for NAFLD (1). Previous Japanese NAFLD patients study also showed a clear relationship between BMI and NAFLD development (30). In our study, mean of body mass index of NAFLD patients excess 25 (29.3 ± 6.1), which indicate overweight. Development and resolution of NAFLD were both closely related to metabolic syndrome especially diabetes mellitus (DM) (30). Hamaguchi et al. reported that 25 - 40% of Japanese NAFLD patients have established DM (3). In our study, 24 cases (30%) of NAFLD patients, had complication of DM. In subgroup, 1 of NAFL (11%) and 23 of NASH (33%), and 16 of NAFLD early stage (30%) and 8 of NAFLD advanced stage (31%) had DM. Because of these demographic features, we regarded that our NAFLD participants could be considered as the representative group of larger population of NAFLD.

d) A description of how participants were recruited 

Reply: We added detail of participants recruiting method as follows. 

(Lines 168-180)

Recruiting and obtaining liver tissue, blood sample and clinical data were performed at Tottori University Hospital and collaborative medical institutes: Hiroshima University Hospital, JA Hiroshima General Hospital, Kawasaki University Hospital, and Shimane University Hospital. from July 2014 to March 2016. All participants were Japanese who live in the western part of Japanese main-island. NAFLD outpatients who underwent continuous clinical follow-up at Tottori University Hospital or collaborative medical institutes were recruited by interview with our study members. In interview, we gave clear exposition of the purpose, procedures, duration and potential adverse events of our study using the printed description which have been obtained the approval of the committee for ethics in medical experiments on human subjects of the medical faculty of Tottori University. Participants were given sufficient time to read the consent and have all questions answered before signing the consent form voluntary.

e) Descriptions of where participants were recruited and where the research took place:

Reply: We described the places of participants recruit and research made.

(Lines 168-172)

Recruiting and obtaining liver tissue, blood sample and clinical data were performed at Tottori University Hospital and collaborative medical institutes: Hiroshima University Hospital, JA Hiroshima General Hospital, Kawasaki University Hospital, and Shimane University Hospital. from July 2014 to March 2016.

(Lines 281-282)

miRNA extract from the serum, Quantitative real-time polymerase chain reaction (RT-PCR) and data analysis were carried out at Tottori University.

In your Methods section, please provide a sample size calculation.

Reply: According to advice of the academic editor, we provide a sample size calculation for our present study in methods paragraph as follows. 

(Lines 156-165)

Sample size calculation

The effective sample size has been calculated from the outcomes of our previous clinical study about serum Dlk1-Dio3 mat miRNA expression in NAFLD patients (25). According to the criterion of our previous study, we set up the relevant difference of miR-379 expression level is 2log2. The associated standard deviation was estimated to be 1.6log2 based on the serum miR-379 expression of normal control in our previous study (25). Given a statistical power of 80% and 0.05 level of significance, a sample size of 11 in each group will be sufficient to detect a clinically relevant difference between groups. To allow for dropout of patients (up to 20%) and further subgroup analysis, we aim to recruit 80 participants.

5. Review Comments to the Author

Reviewer #1: Okamoto and colleagues set out to examine the suitability of circulating serum miR-379 as a potential non-invasive biomarker for non-alcoholic fatty liver disease using real-time PCR platform. The manuscript was well-written and has merit. However, there are some points that need to be clarified.

1. The presence of inhibitors in serum samples may limit the ability to extract RNA or accurately measure miRNAs by real-time PCR. I was wondering whether a spiked-in control was also included in this study to ensure efficiency of RNA extraction? 

Reply: Thank you very much for your advice. We had only examined preliminary study about RT-PCR measurement in human serum between endogenous miR-16 and non-mammal spike in control miRNA. Serum samples were obtained from normal controls (n = 10). 1.6 x 108 copies of C. elegans (Ce)-miR-39-1 (Qiagen, Venlo, Nederland) and 1 μg bacteriophage MS2 RNA (Roche, Penzberg, Germany) as carrier RNA were added for each 200 μL serum sample. Quantitative RT-PCRs was performed using the same protocol of the present study. Both of miR-16 and Ce-miR-39-1 were stably expressed between samples and their standard deviations of threshold cycles were within ± 1 cycle (34.0 ± 0.8 and 23.9 ± 0.1, respectively). We concluded that our miRNA extraction and RT-PCR works steady and endogenous miR-16 could be applied as internal control. However, We did not perform spike in control measurement in NAFLD patients. As the reviewer kindly pointed, we cannot assess equalities in RNA extraction efficacies and RT-PCR accurately measurement between NAFLD serum samples. We disclosed the preliminary study about miR-16 and spike-in control in the materials and methods part. And we also describe the limitation of our study in the discussion part as follows. 

(Lines 300-354 in revised manuscript with track change)

We also examined preliminary study about RT-PCR measurement in human serum between endogenous miR-16 and non-mammal spike in control miRNA. Serum samples were obtained from normal controls (n = 10). 1.6 x 108 copies of C. elegans (Ce)-miR-39-1 (Qiagen) and 1 μg bacteriophage MS2 RNA (Roche, Penzberg, Germany) as carrier RNA were added for each 200 μL serum sample. PT-PCR was performed using the same protocol of the present study. The primer sequence of hsa-miR-16 was UAGCAGCACGUAAAUAUUGGCG. The manufacturer does not disclose the primer sequence of Ce-miR-39-1. Both of miR-16 and Ce-miR-39-1 were stably expressed between samples and their standard deviations of threshold cycles were within ± 1 cycle (34.0 ± 0.8 and 23.9 ± 0.1, respectively). We concluded that endogenous miR-16 could be applied as internal control of RT-PCR in serum miRNA.

(Lines 763-765)

We did not perform spike in control measurement in NAFLD patients. We cannot assess equalities in RNA extraction efficacies and RT-PCR accurately measurement between NAFLD serum samples especially the drop out case.

2. It is not clear how reverse transcription was carried out prior to real-time PCR. Please refer to lines 175-178.

Reply: According to the reviewer’s advice, we added the following sentence to the Materials and Methods section of the manuscript.

(Lines 290-292)

The miScript II reverse transcription kit (Qiagen) was used for reverse transcription of serum miRNA according to the manufacturer’s instructions. 

3. It would be interesting to find out whether non-HDL cholesterol (total cholesterol minus HDL cholesterol) is a predictor for non-alcoholic fatty liver disease. Please refer to Table 1, page 11.

Reply: Thank you very much for your specific advice. According to the reviewer’s advice, we examined the relationships between serum miR-379 expression and non-HDL cholesterol (non-HDL-C) in NAFLD patients. Positive correlation was observed between serum miR-379 expression and non-HDL-C levels in patients with early stage NAFLD. We added the result in abstract, main manuscript, Table 1 and Fig 3. 

4. What was the basis of choosing miR-379 from a series of potential miRNAs (miR-127, miR-136, miR-376c, miR-379, miR-409-3p, miR-411, miR-495) mapped in the 14q32.3 maternally imprinted gene region? Please refer to lines 29-33 and lines 105-111).

Reply: Our explanation for miRNA selection from the miRNA cluster of 14q32.3 maternally imprinted gene region might have been insufficient; therefore, we revised and added the following text. 

(Lines 282-288)

miR-379 was selected from the putative Dlk1-Dio3 mat miRNA cluster because it exhibited the greatest difference in serum expression between NAFL and NASH patients in our preliminary study (25). Comparing to the normal controls, serum miR-379 overexpressed in simple steatosis (we named NAFL to simple steatosis in our previous report) group (3.3 ± 3.1 log2) and down regulated in NASH patients group (-7.4 ± 5.9 log2) in our previous study (25).

Reviewer #2: The manuscript of Okamoto and colleagues highlights the correlation between the expression of miR-379 and NASH development and progression. The study was performed with serum collected from patients affected by early and advanced NASH.

The study was performed in an extremely accurate manner and based its achievement on the correlation between expression of miR-379 and the clinical markers of NASH, especially cholesterol lipotoxicity.

The soundness of the study is supported by previous study published from the same groupd with NASH mouse model.

Some minor points need to be addressed:

The number of patients involved in the study should be corrected between the abstract, the methods and the results section. Please include in the Abstract the number of control samples included in the study. In the Methods section is stated that the patients are 90 and it does not correspond with the abstract. In the Results section, the authors explain that one patient was excluded from the study. This information should be included in the Methods.

Reply: Thank you for your pointing out about our mistake. We corrected patient number; total 80, and indicated the data of excluded patient in abstract and materials and methods part. 

In order to better follow the results described in the text, to add capital letters to differentiate the different panels of the figures when necessary. For example, Figure 1 showing three different graphs of different patient groups.

Reply: Thank you very much for your kind advice. We revised layouts of Fig1, 2 and 3 and adding capital letters. 

Reviewer #3: The authors report increased expression of miR-379 in patients with NAFLD as compared to 10 control patients. They further show that miR-379 may be used as marker of NAFLD with AUROC of 0.76. Computer based prediction of miR-379 targets was preformed and IGF1, IGF1R and INSR were suggested as its targets. These results are of potential interest as they are reporting on potential biomarker of NAFLD in human subjects. Unfortunately, the authors did not adequately describe their results and the Result section appears much shorter than the Introduction. Also, they did not make an attempt to validate downstream targets of miR-379, which is a major missed opportunity.

Major Remarks

It is unclear how many patients were enrolled in the study. The authors state that “Ninety patients were enrolled in this study. The patients were divided into three groups, as follows: 10 patients with asymptomatic gallbladder stones as disease controls, 9 NAFL patients, and 71 NASH patients. In another analysis, NAFLD patients were divided into early stage (n = 53) and advanced-stage (n = 26) groups. (Lane 135-137).” Was the second analysis done on the same set of pts, or was it done on NASH pts only reported in the first analysis. If it was done on NASH pts why are the number different? First cohort 71 NASH pts, while second analysis included 53+26 = 79 patients. To make this clearer, table one should list the number of pts in each group. Furthermore, it should be clearly stated if this was done on one group of pts with two separate analysis or if two cohorts of pts were used.

Reply: Thank you very much for pointing out our mistake. We corrected patient number and indicated the patient numbers in table 1. 

Our 2 NAFLD subgroup studies (NAFL or NASH and NAFLD early stage or advanced stage) were examined same NAFLD patients group (n = 79). We revised the statement of NAFLD subgroup in the materials and methods part. 

(Lines 224-226 in revised manuscript with track change)

The NAFLD patients were divided into two subgroups as follows: 9 NAFL patients, and 70 NASH patients. In another analysis, NAFLD patients were also divided into early stage (n = 53) and advanced stage (n = 26) groups.

I could not readily find figure legends. Some description of figures was found in the Results section but it was short and limited. It did not sufficiently explain the figures, nor did it indicate how many pts were included in each group. Is +/- denoting SE or SD? Figure 2 has only title listed and no description of the figure at all!

Reply: Our figure legends might be insufficient to explain the images correctly. We revised figure legends according to the reviewer’s advice.

It is important to better understand the role of miR-379 in liver physiology. The authors attempted to do this by identifying potential target genes of miR-379 using web-based software. More information needs to be provided about this process before we can make any conclusions about the results that they are showing us.

Reply: Thank you very much for your advice. We added the information about our study design in exploring miR-379 physiological functions. We also added the explanations about the results of GO ontology annotation and GO enrichment analysis. 

(Lines 357-375)

The physiological roles of miR-379 in liver are still hardly identified. Few former studies revealed the relationship between liver function and miR-379. A comprehensive investigation for miR-379 function should be carried out. We adopted the computational miRNA target genes prediction. Software prediction of miRNA target genes is a popular and reliable method to estimate miRNA physiological functions (33). We used web-based software DIANA microT-CDS 5.0 (http://diana.cslab.ece.ntua.gr/) for miR-379 putative target gene analysis. The threshold for the target prediction score in DIANA microT-CDS was set to 0.7. One miRNA can interference hundreds to thousands genes (20). We selected gene ontology (GO) annotation to select NAFLD related genes. Database for Annotation, Visualization, and Integrated Discovery (DAVID) 6.8 (http://david.abcc.ncifcrf.gov/) was used for GO annotation, and the Kyoto Encyclopedia of Genes and Genomes (KEGG) was used for pathway enrichment analysis.

(Lines 503-504)

Simple gene counting of GO terms showed that cellular process, metabolic process and biological regulation had a large proportion amount to over 70%.

(Lines 511-521)

We also examined GO term enrichment analysis. The analysis can identify GO terms, which are significantly over-represented for DAVID pre-built human genome backgrounds (34). miR-379 predicted target genes were richly represented in 12 GO terms compared to DAVID pre-built human genome backgrounds significantly (Table 2). Among 12 enriched GO, 10 terms related to cellular process regulations. 

(Lines 530-535)

Each of the enriched GO terms still had hundreds genes (Table 2). It is one of the common weak points of this method that the output of genes can be large (34). To select putative target gene for NAFLD pathology, we examined extending analysis using backend annotation database. Our GO term analysis showed that miR-379 seemed to relate biological regulations largely (Table 2). Therefore we selected KEGG pathway as the backend database.

The authors found 27 predicted gene targets of miR-379 that were associated with NAFLD development or progression. They focused their discussion on insulin-like growth factor-1 (IGF-1) and IGF-1 receptor. It would be imperative to verify expression of IGF1, IGF1R and INSR in these patients and determine if their expression indeed correlates with expression of miR-379.

Reply: We agree with your opinion. We also recognize it to be the biggest weak point of our present study. However, we do not have enough patient serum and liver tissue to confirm mRNAs and protein expression levels any more. 

We discussed about this problem in limitation part of discussion as follows.

(Lines 774-779)

We did not confirm whether miR-379 actually interfered with any of the predicted target genes in vivo (e. g. expression measurement in serum or liver tissue) or in vitro, such as direct binding experiments or miR-379 ectopic overexpression by gene transfection. Complex intracellular regulatory networks influence the tissue-specific function of miRNAs (115). Therefore, further studies are needed to assess whether the predicted targets are actual targets of miR-379 in NAFLD.

Minor remarks:

That authors state that NAFLD “disease process begins with the development of insulin resistance resulting from excessive energy intake [5]. (Lanes 66-67). While insulin resistance may be universally found in pts with NAFLD, its role in pathogenesis is still unclear. Competing theories exist, with one suggesting that hepatic insulin resistance initiates development of NAFLD, whereas the other argues that insulin resistance is merely a consequence of increased liver lipid deposition. If the authors choose to stand by their wording, a better reference is needed than #5, which mainly talks about the effects of fructose in pathogenesis of NAFLD.

Reply: According to the reviewer’s advice, we added sentences and references for the role of fructose in NAFLD development as follows. 

(Lines 74-85)

This theory suggests that the disease process begins with de novo lipogenesis (DNL) by increase fructose consumption by western style diet and the development of insulin resistance resulting from excessive energy intake (5, 6). Fructose is a potent lipogenic carbohydrate contributing to hepatic steatosis. Fructose is taken into hepatocyte via glucose transporter 2 and converted into fructose-1-phosphate (F1P) by fructokinase. These physiological sequences are not controlled by insulin and induced by fructose (6). Fructose-bisphosphate aldolase B (known as hepatic aldolase) converts F1P into glycogen, glucose, lactate, and acetyl-CoA. Fructose also upregulate key transcription factors for fatty acid synthesis such as sterol response element binding protein 1c and carbohydrate responsive element binding Protein (7). Both acetyl-CoA oversupply and induce of lipogenic enzymes increase DNL in hepatocyte strongly.

---

## [Decision Letter · Decision Letter 1]

11 Feb 2020

Serum miR-379 expression is related to the development and progression of hypercholesterolemia in non-alcoholic fatty liver disease

PONE-D-19-17509R1

Dear Dr. Kinya Okamoto,

We are pleased to inform you that your manuscript has been judged scientifically suitable for publication and will be formally accepted for publication once it complies with all outstanding technical requirements.

With kind regards,

Catherine Mounier

Academic Editor

PLOS ONE

Additional Editor Comments (optional):

Reviewers' comments:

Reviewer's Responses to Questions

**Comments to the Author**

1. If the authors have adequately addressed your comments raised in a previous round of review and you feel that this manuscript is now acceptable for publication, you may indicate that here to bypass the “Comments to the Author” section, enter your conflict of interest statement in the “Confidential to Editor” section, and submit your "Accept" recommendation.

Reviewer #1: All comments have been addressed

Reviewer #2: All comments have been addressed

Reviewer #3: All comments have been addressed

2. Is the manuscript technically sound, and do the data support the conclusions?

Reviewer #1: Yes

Reviewer #2: Yes

Reviewer #3: Yes

3. Has the statistical analysis been performed appropriately and rigorously? 

Reviewer #1: Yes

Reviewer #2: Yes

Reviewer #3: Yes

4. Have the authors made all data underlying the findings in their manuscript fully available?

Reviewer #1: Yes

Reviewer #2: Yes

Reviewer #3: Yes

5. Is the manuscript presented in an intelligible fashion and written in standard English?

Reviewer #1: Yes

Reviewer #2: Yes

Reviewer #3: Yes

6. Review Comments to the Author

Reviewer #1: To the best of my knowledge, all the comments have been carefully and adequately addressed by the authors.

The manuscript has met all the criteria for publication.

Reviewer #2: The authors revised the manuscript according to the reviewers' suggestions. The manuscript quality has consistently improved and, to my opinion, it is suitable for publication.

Reviewer #3: The authors have addressed all my comments successfully. Unfortunately they did not have enough serum to validate expression of IGF-1, IGF1R, and INSR but they did acknowledge that this is the biggest weakness of the manuscript. I do not have additional comments for the authors.

7. PLOS authors have the option to publish the peer review history of their article (what does this mean?). If published, this will include your full peer review and any attached files.

Reviewer #1: Yes: Michael O. Baclig

Reviewer #2: Yes: Pietro Di Fazio

Reviewer #3: Yes: Samir Softic

---

## [Editor Report · Acceptance letter]

14 Feb 2020

PONE-D-19-17509R1 

Serum miR-379 expression is related to the development and progression of hypercholesterolemia in non-alcoholic fatty liver disease 

Dear Dr. Okamoto:

I am pleased to inform you that your manuscript has been deemed suitable for publication in PLOS ONE. Congratulations! Your manuscript is now with our production department. 

With kind regards,

on behalf of

Dr. Catherine Mounier 

Academic Editor

PLOS ONE